# Experimental Study on Tensile Characteristics of Layered Carbonaceous Slate Subject to Water–Rock Interaction and Weathering

Erqiang Li [1], Yanqing Wei [1,*], Zhanyang Chen [2], Paul Archbold [3] and Brian Mullarney [3]

1   School of Civil Engineering, Luoyang Institute of Science and Technology, Luoyang 471023, China
2   School of Civil and Transportation Engineering, Henan University of Urban Construction, Pingdingshan 467036, China
3   School of Engineering & Materials Research Institute, Technological University of the Shannon, N37 HD68 Westmeath, Ireland
*   Correspondence: wyq1987@lit.edu.cn; Tel.: +86-18625799189

**Abstract:** The transverse isotropy of rock masses formed by sedimentation is a common stratum environment in engineering, and the physical–mechanical properties can degrade due to water–rock interaction (WRI) and natural weathering, which potentially lead to the instability or collapse of tunneling, slopes and mining. Taking the carbonaceous slate of the Muzhailing tunnel as the research object, two types of specimens, which include oven-drying (instant drying in oven after fabrication) and natural air-drying (static weathering for 60 days after fabrication) were prepared, respectively, after which Brazilian tests were carried out and the tensile properties were analyzed under the two conditions. The experimental results showed that the two kinds of carbonaceous slate all show brittle failure, but the mechanical response such as failure displacement and peak load is obviously different. The tensile strength of the specimens is significantly all affected by the bedding, while the cleavage failure patterns of the two kinds are affected to different degrees. The softening coefficient of the natural air-drying specimen is 0.11–0.13, which implies that WRI and natural weathering play a vital role in the course of rock failure but have little influence on the transverse isotropy tensile property of bedding. Moreover, the mechanisms of specimen failure subject to WRI and 60 days' weathering were explained by the SEM technique, which analyzed the micro-components and observes the process of specimen deterioration due to physicochemical reaction, the gradual development of cracks and erosion by weathering.

**Keywords:** carbonaceous slate; bedding; tensile strength; failure morphology; water–rock interaction; weathering; mechanism

## 1. Introduction

Various projects such as mine mining, traffic tunnels and hydropower projects are developing vigorously. Most of them involve rock tension failure, and they have attracted considerable interest in geotechnical engineering [1]. Bedding is the most important structural feature of sedimentary rock on the earth, while tensile capacity varies greatly under different bedding conditions. Therefore, it is necessary to focus on the study of bedded rock masses. Among them, slate has obvious bedding characteristics and is widely distributed in China. The Muzhailing tunnel of G75 Lan-Hai (Lanzhou-Haikou) expressway in Min county currently under construction involves the tension failure of slate subject to WRI and natural weathering, so it is of great significance for clarifying its tensile characteristics for guiding engineering practice.

Now many scholars have studied the tensile strength of bedded rock masses such as layered sandstone, schist, gneiss, shale and so on by Brazilian tests. Cho and Dan et al. [2,3] studied the anisotropy of mechanical parameters through uniaxial compression and Brazilian

tests of transverse isotropy gneiss, shale and schist, while they showed that the tensile strength and failure mode are closely related to the bedding. Tavallali and Khanlari et al. [4–6] carried out uniaxial compression and Brazilian tests on layered sandstone, and analyzed the influence of bedding on strength, failure mode and fracture length. Nezhad et al. [7] conducted Brazilian tests and numerical simulation of shale and studied the influence of the bedding and matrix on tensile strength and fracture toughness.

In view of slate, Li et al. [8,9] carried out uniaxial compression and Brazilian tests on layered slate, revealing its strength characteristics and failure mechanism. Debecker et al. [10,11] made experimental research on its failure morphology and the test results are in good agreement with the Udec simulation. García-Fernández [12] studied the shear strength of slate based on the Mohr–Coulomb criterion and Brazilian tests, and the comparison was made by shear test. Moreover, water is the most important environmental factor affecting the mechanical properties of the rock mass, especially the degradation effect of clay-rich minerals rocks [13–21]. Yang and Mao et al. [18,19] carried out triaxial compression tests on slate, analyzed the influence of water absorption changes on the mechanical parameters and summarized the mechanism of WRI through microstructure changes. Gholami et al. [20] studied the influence of bedding and water content on the tensile, compressive strength and elastic modulus of slate through uniaxial, triaxial compression and Brazilian tests under dry and wet conditions. Wang [21] studied the influence of the bedding and water content on the tensile strength by Brazilian tests on dry and saturated slate and showed the water weakening effects.

In conclusion, most of the above works are based on the Brazilian tests of layered rock and common slate, and the effect of WRI on the weakening of slate is obvious, but it usually involves the test conditions of water saturation, drying and wetting cycles and specific water content. However, in the actual engineering such as tunnel excavation, most of them show the natural weathering state of the surrounding rock after soaking for a certain time, while there is a dearth of studies with respect to the effect of natural weathering after WRI. It is therefore of great importance to carry out Brazilian tests of carbonaceous slate subject to WRI and weathering, so as to reveal its tensile characteristics, degradation mechanism and provide a solid theoretical basis and detailed test data for engineering practice, and furthermore provide references and a technical basis for the design and construction of related rock engineering in slate areas with a wide engineering background too.

The outline of this paper is presented as follows. In Section 2, the basic properties of carbonaceous slate, Brazilian disc sampling and Brazilian tests are described. The test results such as the mechanical response, tensile strength and splitting failure morphology of oven-drying and natural air-drying specimens are analyzed in Section 3. Next, the mechanism of the static weathering of carbonaceous slate with specific moisture content under the specimen fabrication process is discussed in detail by SEM in Section 4. Finally, a number of conclusions are drawn in Section 5.

## 2. Brazilian Disc Specimens and Test Setup

### 2.1. Properties of Carbonaceous Slate

The carbonaceous slate was collected from the Muzhailing tunnel of Hai-Lan expressway in Min country, Gansu Province, China (Figure 1). In order to reduce lithological differences, rock blocks were all taken from the same place. This type of carbonaceous slate shows well-defined fissility, has a thin-bedded structure, and is mostly fragmented and loosely structured. The nature of the rock mass brings great difficulties to sampling as failure occurs along weak planes easily, resulting in a very low sampling rate.

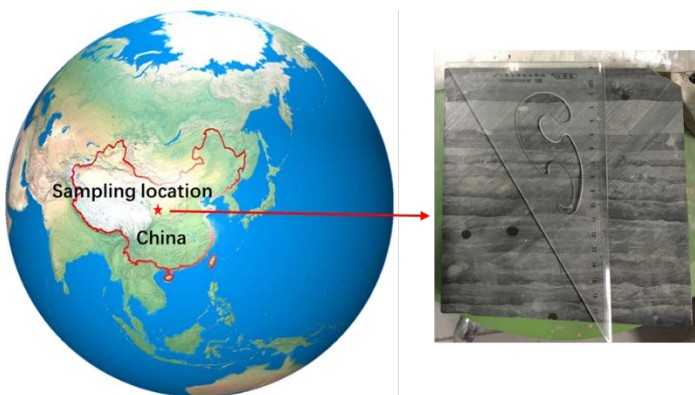

**Figure 1.** Sampling of carbonaceous slate.

The P-wave velocity of parallel and vertical bedded carbonaceous slate is about 3780–4000 m/s and 2940–3360 m/s. The average density of samples processed in the laboratory is 2.688 g/cm$^3$. The uniaxial compressive strength, elastic modulus and Poisson's ratio of horizontal and vertical bedded cylindrical samples are 48.8 MPa and 50.5 MPa, 6.5 GPa and 7.8 GPa, 0.2 and 0.23, respectively. Its mineral composition was determined by means of X-ray diffraction, and mainly composed of quartz (48.8%), clay minerals (47.9%), plagioclase (1.8%), pyrite (0.9%) and potash feldspar (0.6%). The clay minerals are mainly composed of illite, chlorite, illite-smectite formation, kaolinite and smectite.

The microcosmic pore structure of the carbonaceous slate is observed by the casting thin sections in Figure 2. The whole structure is silt-fine-sandy structure and the debris are mainly linear contact. The main compositions of the debris are quartz, feldspar and mica, and particle size of the debris is about 0.01–0.15 mm, and porous cementation is formed. The argillaceous fills the interstitial space partly and is partially aggregated in lumps. Calcite is distributed dispersedly. Punctate iron and a few granular dissolved holes are found in the rocks. There are three structural microcracks, two of which are 0.01 mm wide and not filled, and one is 0.03 mm full-filled with silica.

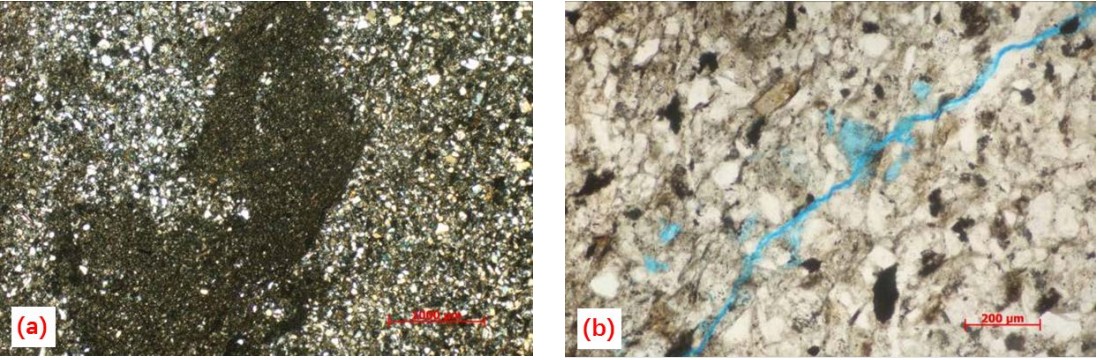

**Figure 2.** Casting thin section inspection: (**a**) fine sandy structure, argillaceous mass and (**b**) microcracks, corrosion hole.

### 2.2. Brazilian Disc Specimens

There are two steps in disc specimen fabrication: (1) Diameter 50 mm sleeve coring and disc cutting under predetermined bedding angles; (2) Polishing discs by M250AH grinder to Brazilian disc specimens with a thickness of approximately 25 mm, in which the plane along the thickness direction should be flattened to less than 0.01 mm, and the concave plane deviates from the plane direction from vertical to a thickness of less than 0.5°, so that the specimens can meet the requirements of the suggested ISRM standard methods [22].

In the process of fabrication, water as lubricant in specimen fabrication is essentially a process of water absorption and initial WRI for carbonaceous slate. The contact state and time between the specimens and water are basically the same, and the time is about 1.5 h. Therefore, the water content of all disc specimens is basically the same after fabrication as the WRI state was all the same due to artificial control. In this paper, the disc specimens were simplified to be transferred to room temperature and natural ventilation for static natural weathering for 60 days (short for air-drying specimen). Meanwhile, another batch of disc specimens was dried in an oven instantly after fabrication and then sealed (short for oven-drying specimen). Through the drying method [23], we concluded that the water content of the specimens is about 1.63–1.95% after fabrication, and the water content of the air-drying specimens is only about 0.21–0.24% after static weathering. The sample preparation process is shown in Figure 3.

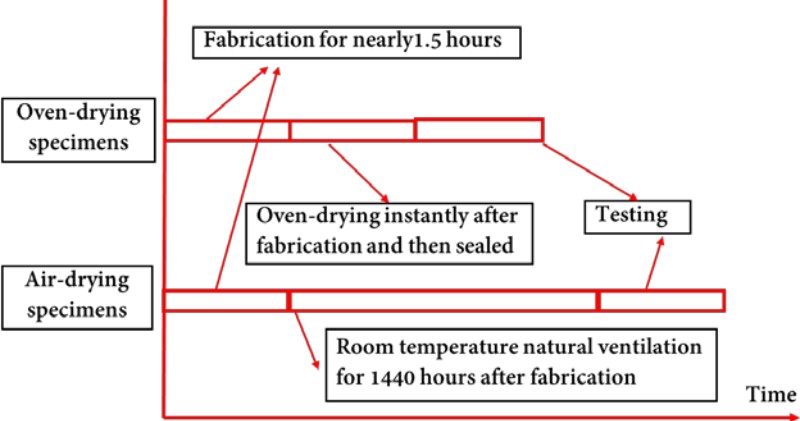

**Figure 3.** Flow chart of carbonaceous slate Brazilian test.

The specimens perpendicular to the transverse isotropic plane have a bedding plane. The angle alpha between the bedding plane and the horizontal plane is taken as the bedding angle, and the alpha is divided as 0°, 30°, 45°, 60°, 90° for the two kinds of carbonaceous slate, and the air-drying and oven-drying specimens are named as SA, SB, SF, SC, SD and IA, IB, IF, IC, ID, respectively, as shown in Figure 4. For example, SA-1 and IF-1 refer to 0° air-drying specimen 1# and 45° oven-drying specimen of 1#, respectively.

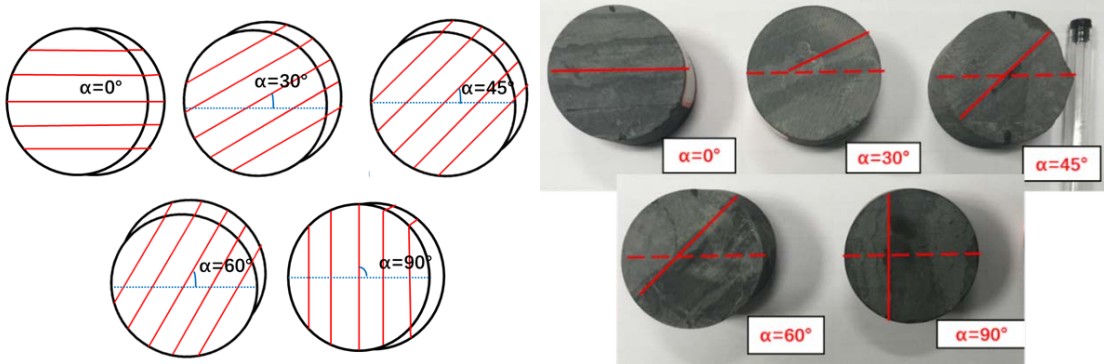

**Figure 4.** Schematics of air-drying Brazilian disc specimens, with α = 0°, 30°, 45°, 60°, 90°.

### 2.3. Brazilian Test Setup

The Brazilian tests of both specimens were all carried out by using the 30 kN photoelastic precision uniaxial compressive equipment of the State Key Laboratory of Geotechnical Mechanics and Underground Engineering (SKLGMUE), as shown in Figure 5. The concentrated line load was applied by using two flat platens with the simplified ISRM

standard [22], managed by the displacement control with a constant rate of loading of 0.1 mm/min. Each group of the oven-drying specimens was carried out three times, and 15 tests were actually conducted successfully, while because of the discreteness caused by continuous WRI and weathering, each group of air-drying specimens was carried out at least 4 times, and 21 tests were actually conducted successfully.

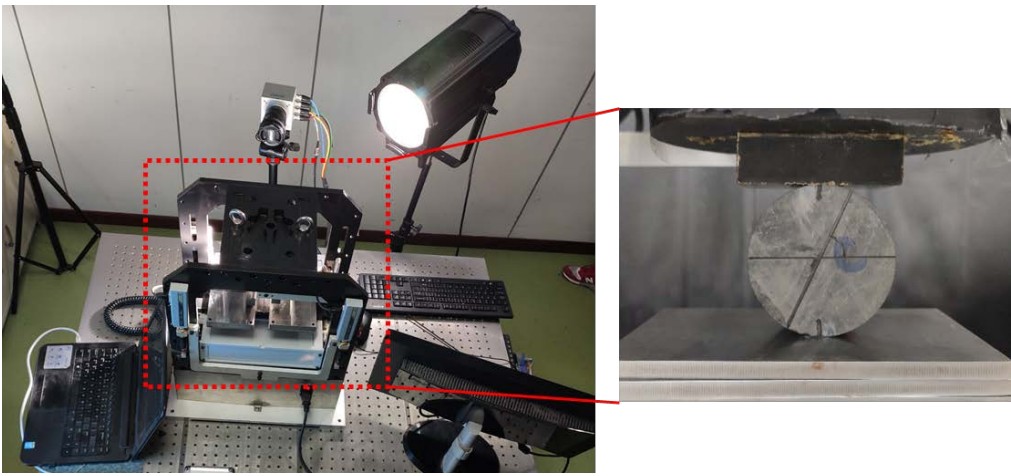

**Figure 5.** Photograph of experimental setup and specimen placement.

## 3. Analysis of Test Results

### 3.1. Mechanical Response

#### 3.1.1. Load-Displacement Curves of Oven-Drying Specimens

The load-displacement curves of all the oven-drying specimens are shown in Figure 6. With the gradual increase in the axial displacement, all curves generally can be divided into two stages as linear elastic and failure drop phases. The curves are a vertical line with a sharp drop after the axial load reaches the peak point, and the specimens produced a load "click" noise and lost its bearing capacity almost in a short time, which conformed to the fracture characteristics of quasi-brittle materials.

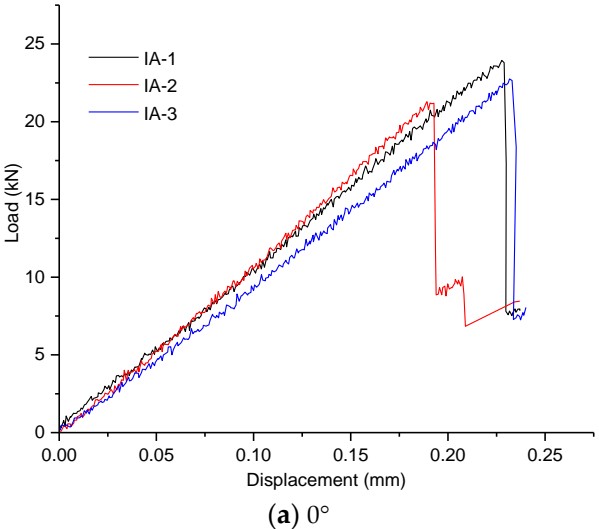
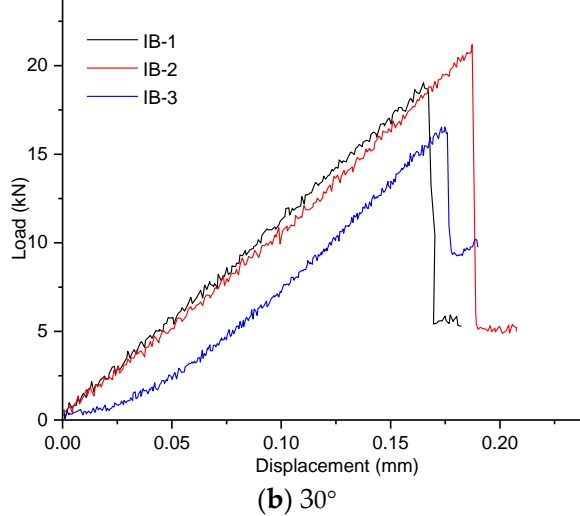

(a) 0°  (b) 30°

**Figure 6.** *Cont.*

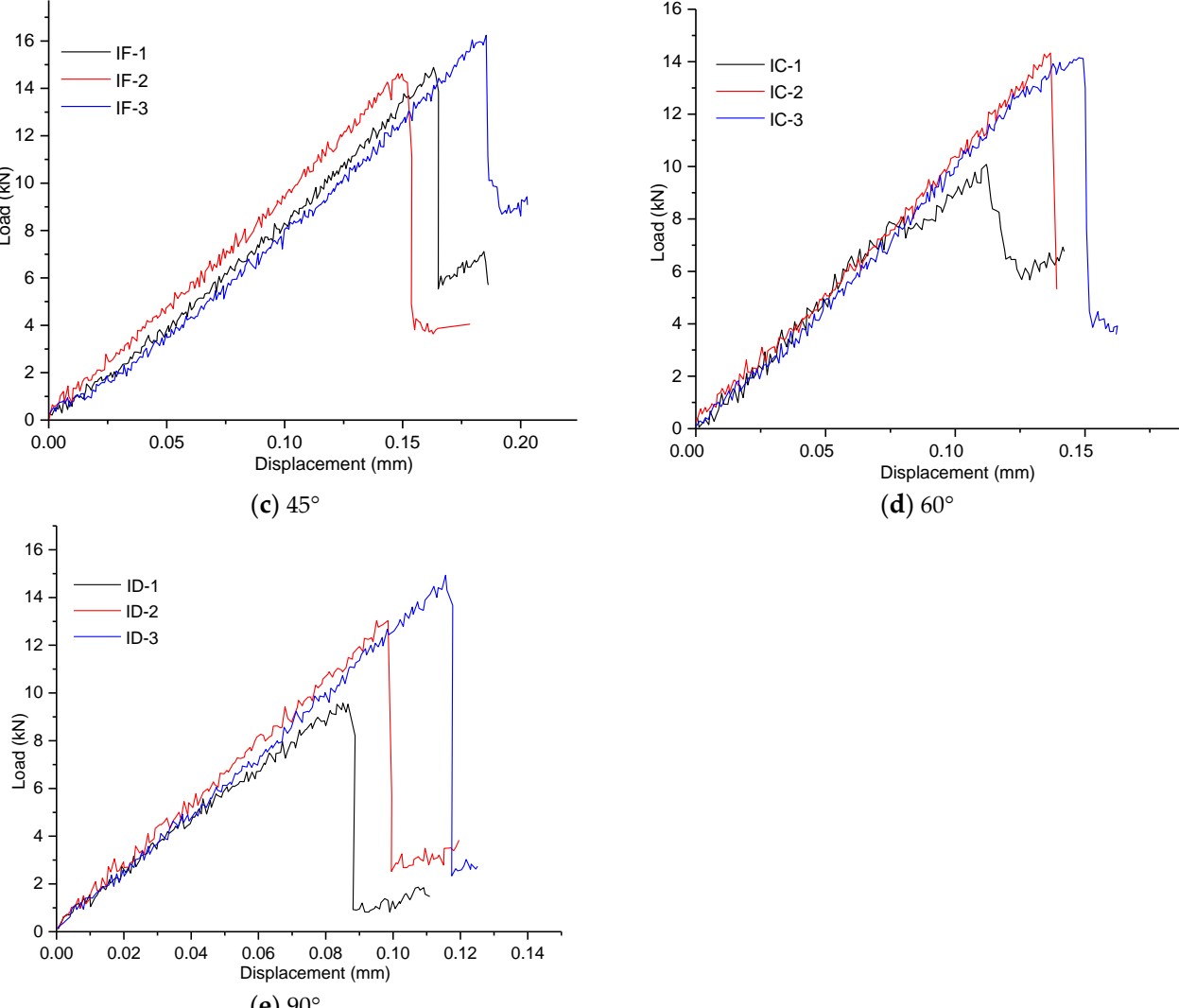

**Figure 6.** Load-displacement curves of oven-drying specimens: (**a**) horizontal bedding, (**b**) bedding angles of 30°, (**c**) bedding angles of 45°, (**d**) bedding angles of 60° and (**e**) vertical bedding.

From Figure 6, we can see that the transverse isotropy formed by the bedding has a great influence on the curves' development. For displacement, the vertical displacement of group IA is about 0.190–0.233 mm, while group ID is only 0.084–0.116 mm, and group IB, IF and IC are 0.166–0.187 mm, 0.148–0.185 mm and 0.112–0.149 mm, respectively. By comparison, we know that the displacement of group IA is the largest when all of them experienced splitting failure, which is caused by the layer by layer compression of the cement between the underlying bedding planes under the vertical load. The peak loads of IA, IB, IF, IC and ID are 21.251–23.928 kN, 16.483–21.180 kN, 14.613–16.187 kN, 10.060–14.284 kN and 9.558–14.903 kN, respectively. The peak loads of IA are the largest, which is the toughening effect of the cement between the bedding planes under the layer by layer compression. It is shown that the bedding has a significant impact on the splitting failure in the Brazilian tests. Meanwhile, due to the individual differences and inherent defects caused by the mineral composition and structure, each group of data presents different degrees of discreteness. On the basis of the oven-drying specimens that are dense with hard crystal particles due to the oven-drying effect, the vibration phenomenon exists obviously in the load-displacement curves due to the vibration effect when the load is close to the maximum bearing load as the small size of the experimental instrument.

### 3.1.2. Load-Displacement Curves of Air-Drying Specimens

The load-displacement curves of the air-drying specimens are shown in Figure 7. As axial displacement increases, the curves generally undergo the development of three phases of compaction, linear elastic and failure drop, and brittle characteristics are presented overall. The load increases with the displacement until a peak value is reached, after which the load drops abruptly, which also belongs to quasi-brittle materials. We must keep in mind that a certain number of invalid samples with great discreteness existed in the test, while only the successful test results are shown in Figure 7.

For displacement, the vertical displacement of group SA is about 0.607–0.705 mm, group SD is only 0.221–0.284 mm, while group SB, SF and SC are 0.366–0.476 mm, 0.41–0.508 mm, 0.342–0.431 mm, respectively. The peak loads of SA, SB, SF, SC and SD are 2.330–3.147 kN, 1.929–2.367 kN, 1.302–1.901 kN, 1.264–1.943 kN and 1.117–1.468 KN, respectively. Compared with the oven-drying specimens, the failure displacement of each group increased considerably. In addition, the bearing load of the air-drying specimens is sharply reduced, but the bedding angles still have a significant impact on it.

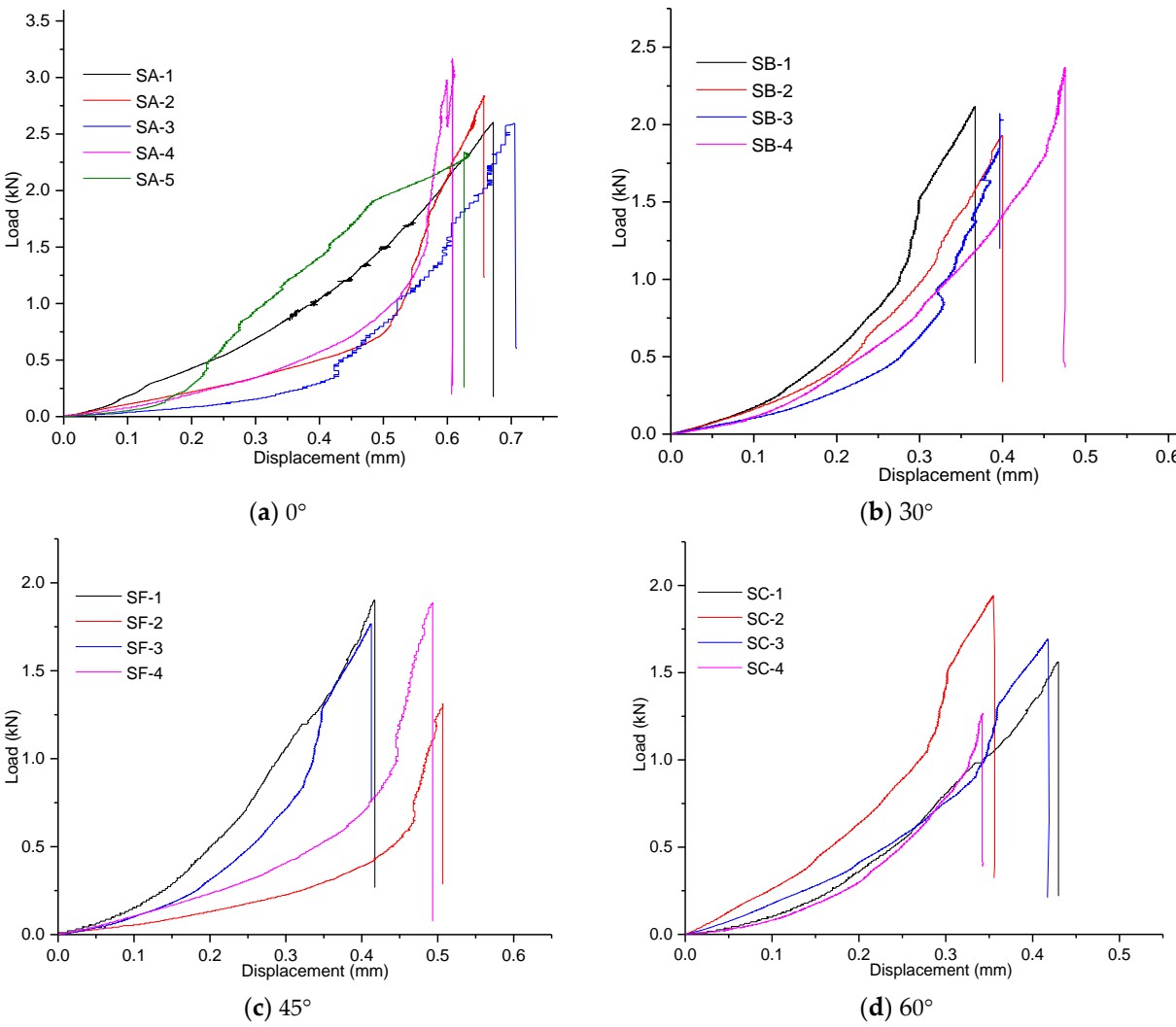

**Figure 7.** *Cont*.

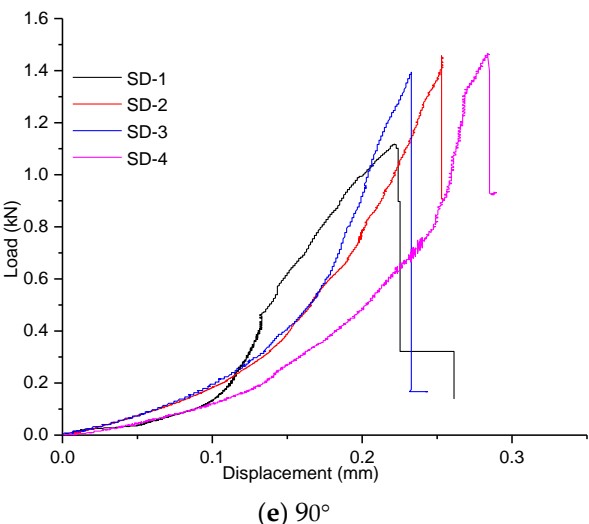

(**e**) 90°

**Figure 7.** Load-displacement curves of air-drying specimens: (**a**) horizontal bedding; (**b**) bedding angles of 30°; (**c**) bedding angles of 45°; (**d**) bedding angles of 60° and (**e**) vertical bedding.

### 3.2. Splitting Failure Morphology

#### 3.2.1. Splitting Failure Morphology of Oven-Drying Specimens

The failure morphology of the oven-drying specimens is shown in Figure 8. The splitting failure modes are divided into three types: central crack, arc crack and bedding crack. Among them, when the bedding angles are 0° and 30°, they were in the form of a central crack, which is near the loading lines at both ends of the crack vertically along the loading line, and the tensile failure characteristics are obvious; when the bedding angle is 45 °, it was in the form of arc crack, which belongs to the tensile-shear failure; while for 60° and 90°, they were in the form of a bedding crack along the bedding direction. The specimens may be tensile or shear failure of the matrix and bedding, a combination of tensile and shear failure of the matrix and bedding, etc., and accordingly show different shapes of the failure cracks. However, in summary, the bedding effect of the oven-drying specimens is significant at the bedding angles of 60° and 90° and is in accordance with reference [24]. It is concluded that under the condition of oven-drying after fabrication, the bonding force of the carbonaceous slate bedding is large, and its overall mechanical properties should be relatively approximate to the matrix.

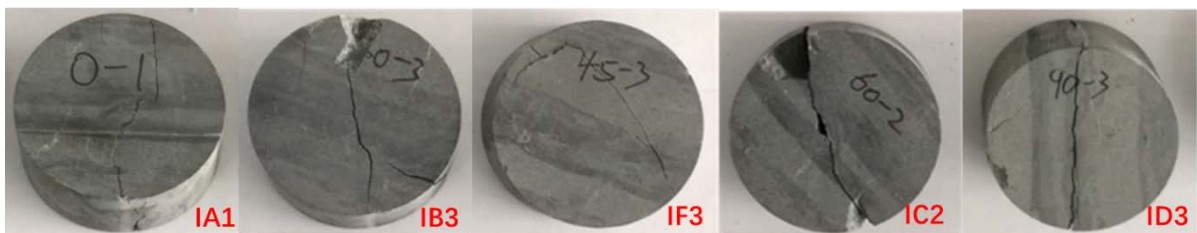

**Figure 8.** Splitting failure morphology of oven-drying specimens.

#### 3.2.2. Splitting Failure Morphology of Air-Drying Specimens

The ultimate failure morphology of the air-drying specimens in the Brazilian tests is shown in Figure 9. The crack path extends along the diameter direction to both ends of the loading and the ends contacted by the loading have shear fractures. The failure morphology can be divided into the following three types: (1) When the bedding angle is 90°, the peak load is generally small. For horizontal tensile stress, the specimen fracture originates from a certain position of the centerline of the discs, and along the bedding the disc is divided into two parts by splitting and pulling. The tensile strength measured by this group can be regarded as the bedding planes' tensile strength; (2) For 30°, 45° and 60°, shear failure

usually occurs along the bedding, which makes the fracture path deviate from the centerline of the vertical samples, such as in the SC (60°) and SF (45°) specimens, where the cracks have easily developed along the bedding plane under shear action. When the bedding angle is small, such as in the SB (30°) specimens, the crack no longer develops along the bedding straightly, but both slip between the beddings and the longitudinal tensile-shear failure of the specimens' matrix, which usually shows a tortuous arc failure morphology; (3) When the bedding angle is 0°, the bedding usually shows the overall toughening effect on the specimens, and the larger peak load will lead to the splitting failure. The concrete manifestation is that when the vertical main crack develops through the centerline, the two ends of the loading will be crushed or sheared. The transverse secondary tension cracks of mineral particles in the matrix are caused by the tensile stress generated by the physical effect, which also results in the complex failure mode of multi-layer cracks. The tensile strength measured by this group can be regarded as the tensile strength of the matrix.

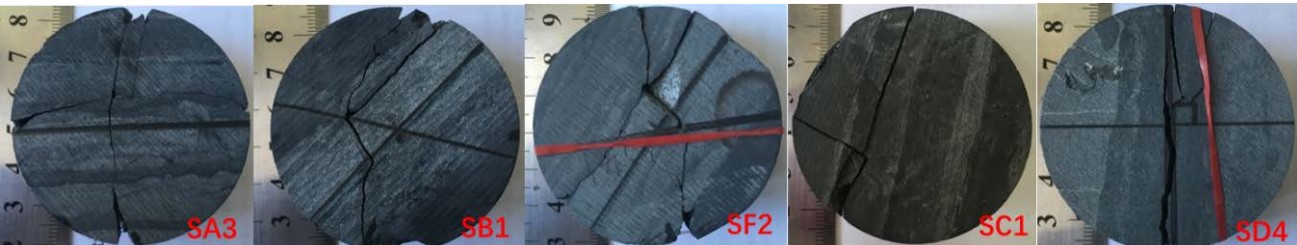

**Figure 9.** Typical failure modes of air-drying specimens under different bedding angles.

Therefore, the anisotropy of air-drying specimens not only affected its tensile strength, but also affected its failure morphology. Compared with the oven-drying specimens, the air-drying specimens have more obvious destruction such as cracks along the bedding, and the splitting failure morphology is actually a comprehensive effect of the mineral matrix, bedding, defects and weathering effect after WRI. Specifically, when the bedding angles approach 0°, the two kinds of specimens are prone to tensile crack along the matrix mainly and may experience shear failure along the bedding plane sometimes; when the bedding angles approach 90°, the two kinds of specimens have tensile cracks along the bedding plane overall.

### *3.3. Tensile Strength*

Based on the assumption of the elastic, homogenous and plane stress condition, the analytical solution of the plane stress [22–25], at any given point such as Q (x, y) of the disc (Figure 10a), is written as Formulas (1)–(3), where the three are normal and the shear stress occurs along the *x*- and *y*-directions, respectively, and *D* is the diameter, *B* is the thickness of the Brazilian samples, and *P* is the applied load at failure.

$$\sigma_x = \frac{2P}{\pi B}\left\{\frac{\left(\left(\frac{D}{2}\right)-y\right)x^2}{\left(\left(\left(\frac{D}{2}\right)-y\right)^2+x^2\right)^2}+\frac{\left(\left(\frac{D}{2}\right)+y\right)x^2}{\left(\left(\left(\frac{D}{2}\right)+y\right)^2+x^2\right)^2}-\frac{1}{D}\right\} \tag{1}$$

$$\sigma_y = \frac{2P}{\pi B}\left\{\frac{\left(\left(\frac{D}{2}\right)-y\right)^3}{\left(\left(\left(\frac{D}{2}\right)-y\right)^2+x^2\right)^2}+\frac{\left(\left(\frac{D}{2}\right)+y\right)^3}{\left(\left(\left(\frac{D}{2}\right)+y\right)^2+x^2\right)^2}-\frac{1}{D}\right\} \tag{2}$$

$$\tau_{xy} = \frac{2P}{\pi B}\left\{ \frac{\left(\left(\frac{D}{2}\right)-y\right)x^2}{\left(\left(\left(\frac{D}{2}\right)-y\right)^2+x^2\right)^2} - \frac{\left(\left(\frac{D}{2}\right)+y\right)x^2}{\left(\left(\left(\frac{D}{2}\right)+y\right)^2+x^2\right)^2} \right\} \tag{3}$$

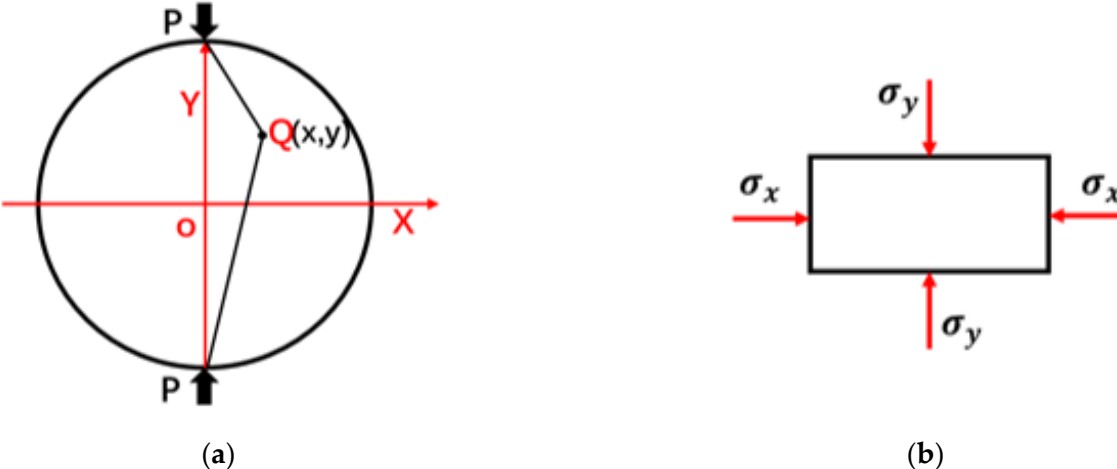

(**a**) 　　　　　　　　　　　　　　　　　　　　　　　(**b**)

**Figure 10.** Schematic of Brazilian test in Cartesian coordinate. (**a**) compressive load of Brazilian test in Cartesian coordinate and (**b**) tensile and compressive stress of central element during Brazilian test.

At the center point of the above sample (Point O in Figure 10a, while the force at point O is shown in Figure 10b), as x = y = 0, the stress state can be written as Formulas (4)–(5), while shear stress is zero. In the x-direction, the tension is set as positive and the compression is negative [24]. Moreover, based on the assumption that the failure occurs at the point of the maximum tensile stress, i.e., at the center of the disc, Formula (4) is used to calculate the Brazilian tensile strength, and we must keep in mind that this may cause some errors as rocks always exhibit some degree of heterogeneity. By using the collected test data, Table 1 lists the calculated tensile strengths, mean value and Standard Deviation (SD) of the two kinds of samples with the different bedding angles.

$$\sigma_x = \frac{2P}{\pi DB} \tag{4}$$

$$\sigma_y = -\frac{6P}{\pi DB} \tag{5}$$

**Table 1.** Statistical data of tensile strength with different bedding angles (Unit: MPa).

| Angles | Oven-Drying Specimens | | | Mean | SD | Air-Drying Specimens | | | | | Mean | SD |
|---|---|---|---|---|---|---|---|---|---|---|---|---|
| 0° | 12.19 | 10.81 | 11.60 | 11.53 | 0.21 | 1.33 | 1.65 | 1.43 | 1.58 | 1.26 | 1.45 | 0.16 |
| 30° | 9.70 | 10.80 | 8.43 | 9.64 | 1.19 | 1.12 | 0.90 | 1.10 | 1.24 | - | 1.09 | 0.14 |
| 45° | 7.45 | 7.56 | 8.17 | 7.73 | 0.47 | 0.97 | 0.73 | 0.85 | 1.01 | - | 0.89 | 0.13 |
| 60° | 5.13 | 7.30 | 7.18 | 6.54 | 1.22 | 0.76 | 0.95 | 0.77 | 0.65 | - | 0.78 | 0.12 |
| 90° | 4.88 | 6.57 | 7.61 | 6.35 | 1.38 | 0.59 | 0.71 | 0.68 | 0.78 | - | 0.69 | 0.08 |

### 3.3.1. Tensile Strength of Oven-Drying Specimens

As shown in Figure 11a, the fitting curve of the oven-drying specimens' tensile strength is obtained. For the bedding angles of 0°, 30°, 45°, 60° and 90°, the mean tensile strength is 11.53, 9.64, 7.73, 6.54 and 6.35 MPa, respectively. When the bedding angle is 30°, 60° and 90°, the dispersion of the test results is significant, which is related to the different cleavage failure patterns of the bedding effect in the test. The calculated anisotropy coefficient of the oven-drying specimens' tensile strength is 1.82, which shows the significant anisotropy

affected by the bedding. From Figure 11, we know that the tensile strength of carbonaceous slate shows a nearly linear decreasing trend when the bedding angles increase gradually, and its change trend of strength is similar to that of many kinds of bedded rocks [2–12].

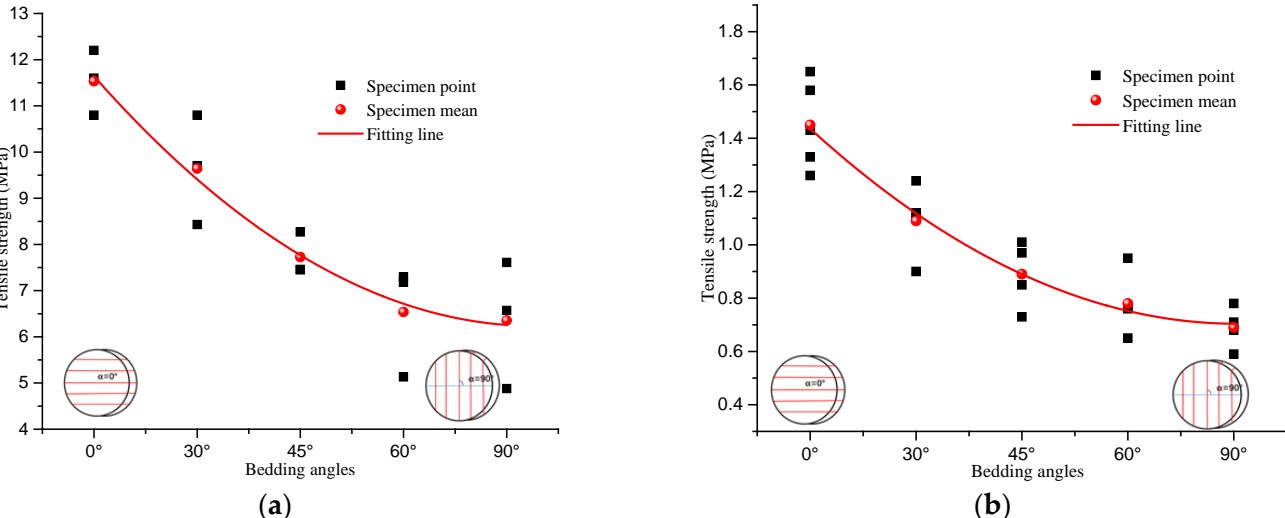

**Figure 11.** Fitting curve of two kinds of specimens' tensile strength at different bedding angles. (**a**) oven-drying specimens and (**b**) air-drying specimens.

### 3.3.2. Tensile Strength of Air-Drying Specimens

The fitting curve of tensile strength for the air-drying specimens is shown in Figure 11b. The mean tensile strength is 1.45, 1.09, 0.89, 0.78 and 0.69 MPa at the bedding angles of 0°, 30°, 45°, 60° and 90°, respectively. The tensile strength is maximum when the bedding angle is 0°, and gradually decreases with the increase in the bedding angle; when the bedding angle is 90°, it decreases to a minimum of 0.69 MPa, and the calculated anisotropy coefficient of the air-drying specimens is 2.10, which fully reflects the typical anisotropy of the tensile strength affected by the bedding. Its strength change trend is similar to the above instant samples and a variety of layered rocks [5–22]. Compared with the oven-drying specimens, the difference in the tensile strength between them in the same bedding state is great, which is caused by the effect of the WRI and long-time natural weathering.

### 3.4. Degradation Effect of Natural Weathering under WRI

Combined with the above tensile strength data, we know that the effect of natural weathering after the process of water lubrication, i.e., WRI, is significant, and the comparison is shown in Figure 12. Referring to the softening coefficient of the hydraulic properties of the rocks, the softening coefficient of the tensile strength by the WRI's natural weathering effect is defined as the ratio of the tensile strength of the rock after a specific natural weathering time at a specific water content to that of the rock treated by air-drying. Comparing the results of the two types of tests, we found that the softening coefficient is only 0.13, 0.11, 0.12, 0.12 and 0.11, respectively, which shows strong softening and poor engineering mechanical properties. This is consistent with the short-term stability after the construction of the Muzhailing tunnel and the serious deformation and failure of the tunnel support after a certain time.

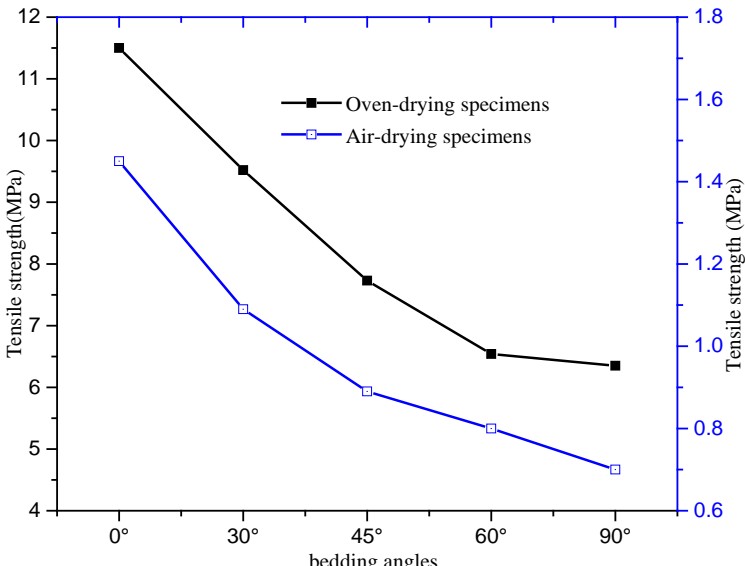

**Figure 12.** Contrast curve of two specimens' tensile strength under different bedding angles.

## 4. Mechanism of Natural Weathering after WRI

The rock is composed of mineral particles, in which there are pores, cracks, defects and microstructural planes. In the process of specimen fabrication, water-rock contact will produce essentially WRI, which will lead to the continuous change of mineral composition and structure under the natural weathering, and then lead to the gradual deterioration of the physical and mechanical properties of carbonaceous slate. This change can be macroscopically reflected by the great difference in color of the slate specimens in Figures 8 and 9. It is a macro-mechanical property deterioration process caused by the change of the microstructure [14–22,25–32]. The degradation mechanism was analyzed by scanning electron microscopy (SEM). The SEM photos of the typical carbonaceous slate specimens are shown in Figures 13 and 14.

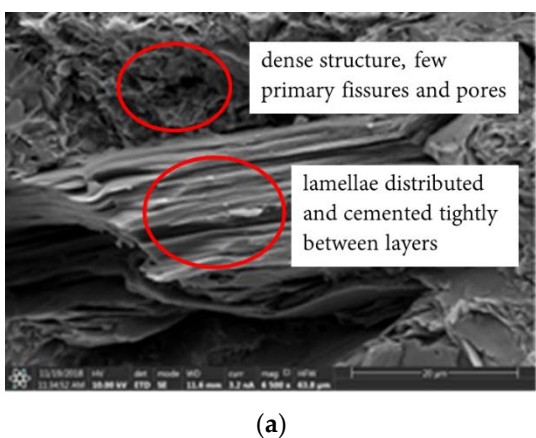

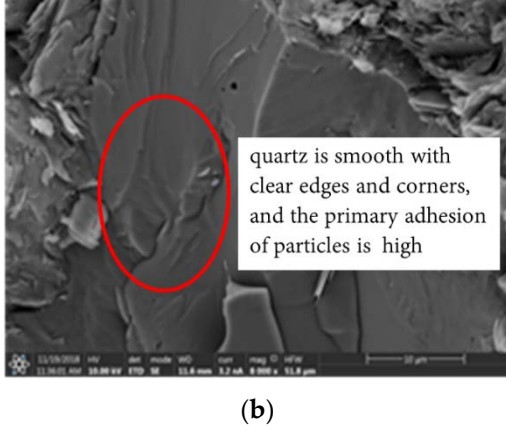

(**a**)

(**b**)

**Figure 13.** SEM micrographs of instant oven-drying carbonaceous slate specimens. (**a**) Specimen I-1#(mag 6500×); (**b**) Specimen I-2#(mag 8000×).

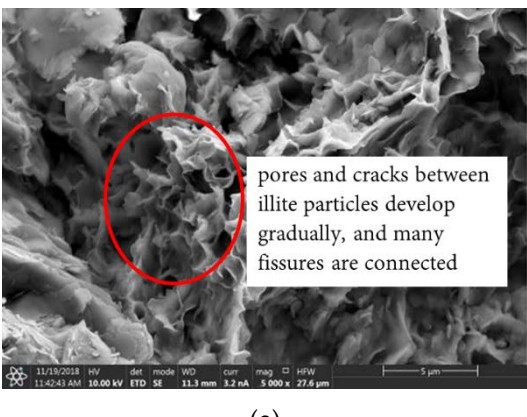
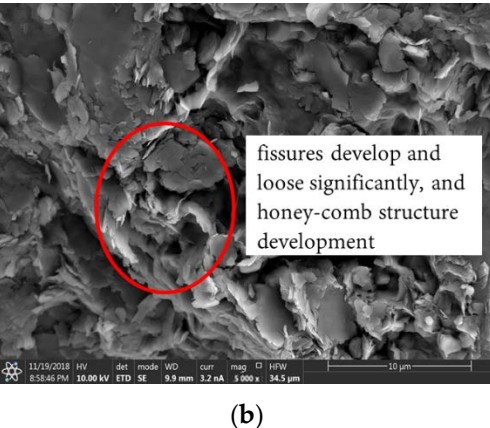

(**a**)                                                    (**b**)

**Figure 14.** SEM micrographs of air-drying carbonaceous slate samples. (**a**) Sample S-1#(mag 5000×); (**b**) Sample S-2#(mag 5000×).

The cross-section of oven-drying samples has clear mineral particles, dense structure, obvious irregular particle shape, few primary fractures and pores, and its size is only between 0–1 μm under the condition of the pores. As shown in Figure 13a, mica particles are distributed in lamellae and cemented tightly between the layers. In Figure 13b, the surface of the massive quartz is smooth with clear edges and corners, and the primary adhesion of the particles is very high as the structure is extremely dense. As the macroscopic behavior of the material usually develops as a result of the evolution of the micro-scale damage, from the above typical microstructure it is inferred that its macro-mechanical properties should be good and are in accordance with the load-displacement of the Brazilian tests, which have the ability to bear the large load and to vibrate easily as a result of the hardness and high bond strength of the carbonaceous slate particles.

Under the air-drying (natural weathering) of the specimens after fabrication, when the initial state of the water content of the air-drying specimen is about 1.63–1.95%, the cement dissolves significantly with the specimen in the air-drying condition. Consequently, fine pores gradually appear among the particles, microcracks develop, and the mutual connection between the particles is weakened due to dissolution, etc. At that point, the particles will produce a material falling off, the edge is gradually blurred, the pores and cracks between the particles develop and connect, the connection between the mineral particles is gradually loosened, and the time effect of the weathering gradually accumulates, resulting in the convergence and penetration of the crack and the scale increasing obviously. In Figure 14a, the pores and cracks between the illite particles develop gradually, and many pores are connected. In Figure 14b, the chlorite cracks develop significantly, the structure is gradually loosened and presents a honey-comb structure development.

From Figure 15a, we know that the carbonaceous slate is ideally composed of mineral particles and adhesion between the particles, pores, microcracks and water vapor. By simplification, its adhesion is composed of the original cohesion $F_1$, the cementation cohesion $F_2$, the surface friction $F_3$ and the gripping resistance $F_4$ between the particles as shown in Figure 15b. The water content of the carbonaceous slate is as high as 1.63% to 1.95% after fabrication because of its strong hydrophilicity as rich in clay minerals. Under the influence of water, the hydration and expansion characteristics of illite, kaolinite, chlorite and other clay minerals will make the mineral particles full of water molecules, and the pore and microstructure may change with the passing of time as natural weathering. For example, the reaction of Formula (6) between illite and water will weaken the connection between the crystal layers, and then reduce the cohesion, resulting in loose intergranular cementation and even the argillization effect. In the weak acid environment formed by tap water and combined with $CO_2$, the reaction of Formulas (7) and (8) of mica, potassium and feldspar shows that the dissolution with the aqueous solution will lead to ion dissolution, migration or a continuous material exchange and chemical reaction. The resulting reactants

are susceptible to metamorphism, decomposition, migration and loss, which will also make the ore minerals loose and easy to be broken by natural weathering effects, which can be collectively referred as physical and chemical effects. As shown in Figure 15c, under the effect of water, the dissolution and other effects of the mineral particles will lead to particle reduction, contact surface reduction, the particle surface becoming round and smooth, and the poor mechanical properties of the adherents on the particle surface.

$$K_{0.9}Al_{2.9}Si_{3.1}O_8(OH)_2 + n\ H_2O \rightarrow K_{0.9}Al_{2.9}Si_{3.1}O_8(OH)_2 \cdot n\ H_2O \tag{6}$$

$$KAl_3Si_3O_{10}(OH)_2 + 10\ H^+ \rightarrow 3\ H_4SiO_4 + 4\ K^+ + 3\ Al^+ \tag{7}$$

$$4\ KAl_3Si_3O_8 + 8\ H_2O \rightarrow 4\ KOH + 4\ Al_4[Si_4O_{10}](OH)_8 + 4\ SiO_2 \tag{8}$$

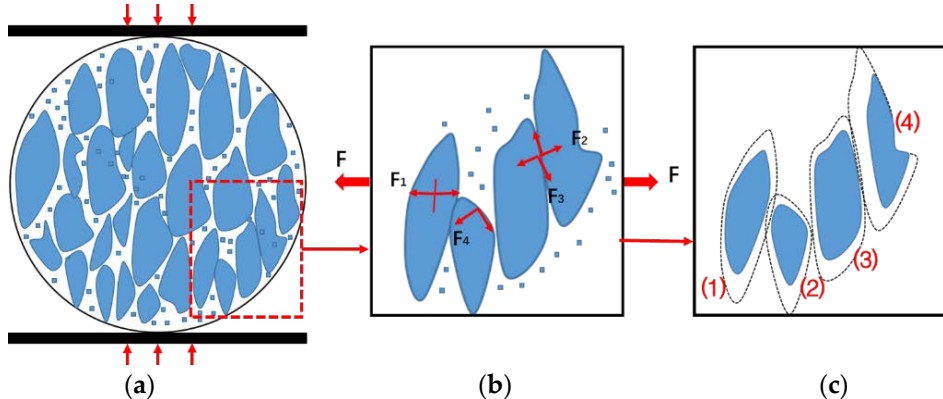

**Figure 15.** Schematic diagram of stress model of carbonaceous slate in Brazilian test: (**a**) Brazil splitting loading diagram, (**b**) schematic diagram of local stress, (**c**) deterioration evolution of mineral particles.

Under the above physicochemical action, the microstructure and layer cementation of the air-drying samples undergo continuous structural adjustment and damage degradation. The structural adjustment is manifested by the dissolution and loss of the mineral components, the increase in pores and defects, the weakening of the cementation among the mineral particles, and the formation of more microcracks in the sample. Meanwhile, as the water gradually evaporates, the natural weathering will also have the degradation effect of pulverization on the mineral particles. As shown in Figure 15c, the shrinkage of the mineral particles occurs again with the evaporation of water, which makes the original cohesion of the mineral particles drop suddenly, and the reduction in surface binder makes the cementation cohesion drop. As the surface between the particles become smooth and the pores between the particles increase, the occlusion resistance and surface friction decrease, which is the mechanical effect presented from the perspective of the micro-particles. It should be noted that the WRI and subsequent natural weathering effect of the specimens are produced after fabrication, resulting in cumulative damage degradation effects, such as the physicochemical reaction under the gradual evaporation of water, the deterioration of the mineral composition and structure and the gradual development of cracks, weathering and erosion of the air-drying specimens. The calculated softening coefficient is only 0.11–0.13, which is a typical quantitative proof. Thus, WRI and natural weathering are the driving forces for the evolution of near surface environment. Under the coupling effect of the physical and chemical evolution, the groundwater solute migration and the change of stratum geological structure in the water-rock system greatly affect the stability of the stratum structure and the evolution of the groundwater environment.

## 5. Conclusions

Brazilian tests were carried out on two kinds of carbonaceous slate with different bedding angles. The development trend of the tensile strength of natural weathering effects under specific water content was studied. The conclusions are as follows:

1.  The load-displacement curves of oven-drying specimens can be divided into linear elastic and failure drop phases, while the air-drying specimens develop from a process of compaction, linear elastic and failure drop phases, and great difference in failure displacement and load from the two kinds of carbonaceous slate specimens.
2.  For oven-drying specimens, the failure mode is less affected by the bedding, and the failure morphology extends nearly symmetrically along the loading axis. However, the failure morphology is all significantly affected by the bedding for the air-drying specimens, and can be divided into pure tension failure, combined tension-shear failure between the matrix and the bedding plane under the different bedding angles.
3.  The magnitude of the peak load is all related to the bedding. With the increase in the bedding angles, the corresponding tensile strength all gradually decreases, in turn as 1.45, 1.09 and 0.89, 0.78, 0.69 MPa for the air-drying specimens, while they are 11.50, 9.52, 7.73, 6.54, 6.35 MPa for the oven-drying specimens, respectively. The softening coefficients are 0.13, 0.11, 0.12, 0.12 and 0.11, respectively. It can be concluded that the clay-rich minerals are vulnerable to WRI and the degradation effect caused by natural weathering.
4.  The degradation mechanism of WRI caused by the fabrication process and the subsequent natural weathering is such that the physicochemical reaction under the gradual evaporation of water, such as the deterioration of the mineral composition and its structure, the gradual development of microcracks and other cumulative damage degradation, effects the weakening of the mineral particles' mechanics. The degradation of microstructure is reflected in the macroscopic mechanical response by our parallel experiments and SEM.

**Author Contributions:** E.L. was responsible for the overall design of the experiment and the writing of manuscripts; Y.W. was responsible for the experimental funding, experimental guidance and revision of the manuscript; Z.C., P.A. and B.M. were responsible for the experiments, data collection and data analysis. All authors have read and agreed to the published version of the manuscript.

**Funding:** This research was funded by the National Key R&D Program of China (No. 2016YFC0600901), the Key research Project of higher education institutions in Henan Province (Grant No. 23B560011) and the Key Science and Technology Research Projects of Henan Province (Grant No. 222102320376). Besides, it was funded by Research Fund for high level talents of Luoyang Institute of Science and Technology (No. 21010714 and No. 21010625).

**Institutional Review Board Statement:** Not applicable.

**Informed Consent Statement:** Not applicable.

**Data Availability Statement:** Not applicable.

**Acknowledgments:** We would like to thank Jili Feng, Yingjun Li and Dejian Li for their kind help on testing suggestions and professional discussions, and associate professor Zhigang Tao and laboratory technician Zhenqun Qi for their kind help on field sampling and samples preparation. At the same time, we need to thank Lu Chen, Mingyuan Zhang, Chao Han, Chunxiao Li and Jiashu Wang for their help during the experiments. Last but not least, the authors would also like to thank Zhenyu Zhang and Yanwei Chen for their assistance in the technical writing of this paper.

**Conflicts of Interest:** The authors declared that no conflict of interest in this work.

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
