# Peer review of "Experimental Study on Tensile Characteristics of Layered Carbonaceous Slate Subject to Water–Rock Interaction and Weathering"

_sustainability, doi:10.3390/su15010885_

Round 1
Reviewer 1 Report
Dear Authors,
My apologies for the delay in the review,
Before publication, a reviewer with more experience in this field must revise the paper because I am not specialist in this field.
Some recommendations:
Contractions are not correct in formal/scientific writing, please revise all. The text must be written in formal English…Revise all.
The figures 6 and 7 must be in the same page. Add A), B)...for better understanding the different graphics. Explain each graph a bit in the figure caption. The quality of structure and clarity must be improved.
Check Fig. X (with space) or Fig.x (without space)
The conclusions do not have points and the phrases are extremely long. Please, revise and replace ; by .
Other comments or corrections are highlighted and commented in the attached PDF.
Good luck and Merry Christmas

Reviewer 2 Report
Regarding the work "Experimental study of the tractional characteristics of stratified carbonaceous slates subjected to water-rock interaction and weathering". The following should be considered:
The summary must have the results, and describe between which ranges the mechanical properties and traction are affected, since this would represent a greater attraction to the article.
The methodology of the characterization procedure and determination of material properties is not described.
Figures 6 and 7 are out of margin
Figure 12 Image quality should be improved
The author mentions in lines 358 to 368 that there is a continuous change of mineral composition and structure under natural weathering, the author is recommended to delve more into the physical-chemical mechanism that is happening in the sample manufacturing process, when there is water-rock contact and enough WRI occurs. Therefore, it is advisable to know the stratigraphic environment to better understand its evolution, such as those mentioned below:
Indeed, from line 413 to 418 "With CO2, the reaction of formula (7) and formula (8) of mica, potassium feldspar, it can be observed that dissolution with aqueous solution will lead to dissolution of ions, migration or continuous exchange of material and chemical reaction, and the resulting reagents are easy to metamorphism, decomposition, migration and loss, which will also make the minerals loose and easy to break by the effect of natural weathering, such as the above can be collectively called effects physical and chemical. As shown in Figure 16c.
It is recommended to further explain the nature of the source of the fluids and convincingly explain fluid migration and water-rock interaction and weathering.
It is recommended that the author present EDS of figures 13 and 14 to justify that it is indeed illite and chlorite, since indeed the honeycomb shape is characteristic of a clay.
The author is recommended to carry out XRD to determine the mineralogical phases of the slate before and after treatment.
It is recommended to update the references.
